# Zinc Borate Hydrolysis

**DOI:** 10.3390/molecules27185768

**Published:** 2022-09-06

**Authors:** David M. Schubert

**Affiliations:** Department of Chemistry and Biochemistry, Metropolitan State University of Denver, Denver, CO 80217, USA; dschube1@msudenver.edu

**Keywords:** solubility, boric acid, zinc oxide, zinc hydroxide, wulfingite, composite materials, preservative

## Abstract

The crystalline zinc borate phase ZnB_3_O_4_(OH)_3_, known in commerce as 2ZnO·3B_2_O_3_·3.5H_2_O, is an important industrial material used as a fire-retardant synergist in polymers, a source of micronutrients in agriculture, and a preservative in building materials. It lends durability to wood composite building materials by inhibiting attack by wood destroying organisms. The hydrolysis chemistry of this zinc borate is relevant to its industrial use. ZnB_3_O_4_(OH)_3_ exhibits incongruent solubility, reversibly hydrolyzing at neutral pH to insoluble Zn(OH)_2_ and soluble B(OH)_3_. It is sparingly soluble with a room temperature solubility of 0.270 wt% in terms of its equivalent oxide components in solution, comprising 0.0267 wt% B_2_O_3_ and 0.003 wt% ZnO. Aspects of the hydrolysis chemistry of zinc borate under neutral pH conditions are discussed.

## 1. Introduction

Zinc borate ranks among the top ten boron-containing chemicals in terms of global production and use. Of the several known zinc borate phases, ZnB_3_O_4_(OH)_3_, having the resolved oxide formula 2ZnO·3B_2_O_3_·3H_2_O, is by far the most commercially important and has been in industrial scale production for more than half a century [1]. Tens of thousands of tons of this compound are used annually in industrial applications, where it is known as an article of commerce as 2ZnO·3B_2_O_3_·3.5H_2_O. This slightly incorrect commercial formula is a result of an early error in characterization, but it is correctly formulated as the trihydrate for its oxide formula [2]. The compound actually contains no free water of hydration, nor free ZnO and B_2_O_3_, which are abstract components of the oxide formula conventionwhich factors out hydroxyl groups as apparent water. While the minor inaccuracy in the commercial formula has no bearing on the properties or efficacy of this compound, the more accurate and concise formula ZnB_3_O_4_(OH)_3_ is primarily used herein.

This is the only zinc borate compound that currently carries biocidal registrations, including with the U.S. EPA and the Canadian PMRA, for certain brands. It is used extensively as a fire retardant synergist in polymers and to lend durability to engineered wood building materials, wood-plastic composites, and related products. In polymers, it is used in combination with other additives to reduce flammability, suppress smoke, and improve electrical properties [3,4]. In building materials, it provides protection against attack by decay fungi and boring insects. In particular, oriented strand board (OSB) siding and sheathing products containing zinc borate preservative have a long record of excellent durability in service extending back more than 30 years [5,6,7,8]. The hydrolysis chemistry of zinc borate has considerable relevance to its industrial applications.

Boric acid is well known to inhibit wood destroying organisms and is a registered biocide in many regions, including USA, Canada, and the EU. However, due to its relatively high water solubility and low dehydration onset temperature (<100 °C), it is often impractical to incorporate boric acid directly into polymers and composite building materials, especially when heat is applied during manufacture. On the other hand, borates that are completely insoluble are not expected to be efficacious as biocides since it is necessary for some amount of borate to be mobile and biologically available in order to control undesirable organisms. Zinc borate exhibits both low solubility and a high dehydration onset temperature, ca. 290 °C, making it suitable for use in many manufacturing processes. Despite its industrial use in many applications for more than half a century, we are unaware of any detailed reports on the hydrolysis chemistry of zinc borate. Herein, we describe measurements of the solubility of zinc borate ZnB_3_O_4_(OH)_3_ and its mode of hydrolysis, which provides a likely mechanism for its efficacy in enhancing durability with respect to biodegradation.

## 2. Materials and Methods

### 2.1. Materials and Measurements

Zinc borate used in this study was a commercially available product produced by U.S. Borax Inc. with a mean particle size of about 5 μm. Powder X-ray diffraction (PXRD) data were collected using a Panalytical Empyrean diffractometer using Cu Kα radiation. Boron and zinc concentrations were measured using a ThermoFisher iCAP 6500 inductively couple plasma optical emission spectrometer (ICP-OES).

### 2.2. Hydrolysis Procedure

Varying weighed amounts of zinc borate were added to a 20-L cylindrical glass battery jar containing measured amounts of deionized water. The container was covered to reduce evaporation, and the resulting mixtures were stirred using a magnetic spin bar at a rate sufficient to keep the zinc borate suspended at room temperature for 72 h. The slurries were then filtered by siphoning the suspension from the tank through a Bückner funnel fitted with medium filter paper, resulting in a completely clear filtrate in the receiving flask. The recovered solids were air dried and submitted to PXRD analysis for identification. Boron and zinc concentrations of the clear filtrate solutions were measured by ICP-OES. Controls with no added zinc borate were undertaken to confirm that any boron that might leach from the borosilicate glass vessel under the experimental conditions was insignificant.

As an example, a 0.05% suspension was prepared by adding 9.00 g zinc borate to 17,991 g deionized water in a glass tank. The suspension was stirred for 72 h and then filtered, producing in a clear filtrate in the receiving flask. The filtered solids were subjected to identification by PXRD. The clear filtrate was submitted to ICP-OES analysis for B and Zn. A portion of the clear filtered was evaporated down at room temperature and resulting solid evaporite submitted to PXRD analysis to identify the solute.

### 2.3. Rate of Dissolution Procedure

Suspensions of zinc borate of known concentrations were prepared gravimetrically in Erlenmeyer flasks and agitated continuously at room temperature at a rate sufficient to maintain the solids in suspension using an orbital stir platform. Portions of the suspensions were removed periodically by syringe, filtered through 5 μm disposable Millipore filters, and the resulting clear filtrates were submitted to ICP-OES analysis for measurement of boron and zinc concentrations.

## 3. Results and Discussion

Zinc borate is produced by the reaction of zinc oxide with excess boric acid at temperatures above 70 °C in aqueous suspension, usually in the presence of product seed, as shown in Equation (1) [9]. If the mother liquor does not contain a sufficient concentration of excess of boric acid, or if the temperature is not sufficiently high, other zinc borate phase may form, such as Zn[B_3_O_3_(OH)_5_]·H_2_O (2ZnO·3B_2_O_3_·7H_2_O) or 3ZnO·5B_2_O_3_·14H_2_O. The structure and chemical formula of ZnB_3_O_4_(OH)_3_ is well defined, based on its single crystal X-ray diffraction structure, shown schematically in Figure 1 [2]. It is an inoborate consisting of chains of triborate rings with interstitial tetrahedral Zn^2+^ cations coordinated to hydroxyl and boroxyl oxygen atoms. The structure is further interconnected by hydrogen bonds. Inoborates generally exhibit relatively slow dissolution rates compared to nesoborates which contain insular borate anions. The structural chemistry of metal borates has been reviewed [10]. Although the formula 2ZnO·3B_2_O_3_·3.5H_2_O used in commerce was shown long ago to be incorrect and properly should be given as 2ZnO·3B_2_O_3_·3H_2_O, these two formulas have different CAS numbers, causing some confusion in the regulatory domain.
excess B(OH)_3_
ZnO + 3 B(OH)_3_  →  ZnB_3_O_4_(OH)_3_ + 3 H_2_O
>70 °C(1)

When zinc borate is added to water at neutral pH in the absence of excess boric acid it exhibits incongruent solubility, hydrolyzing to less soluble zinc hydroxide and a more soluble boric acid, according to Equation (2). Hydrolysis at room temperature proceeds until the boron concentration of the supernatant solution reaches a concentration of about 830 ppm B (77 mM B). The zinc concentration in solution at equilibrium is about 25 ppm (0.38 mM Zn), owing to the much lower solubility of zinc hydroxide. A boron concentration of 830 ppm is equivalent to a 0.475 wt% solution of boric acid. The solution pH remained near 7 during hydrolysis. This can be regarded as a buffered system.
ZnB_3_O_4_(OH)_3_ + 4 H_2_O ⇌ Zn(OH)_2_ + 3 B(OH)_3_
(2)

Solubilities of borate compounds are traditionally expressed as the sum of the anhydrous components of their oxide formulas in solution. Therefore, the solubility of ZnB_3_O_4_(OH)_3_, with oxide formula 2ZnO·3B_2_O_3_·3H_2_O, is expressed as the sum of the equivalent B_2_O_3_ and ZnO concentrations calculated from measured boron and zinc concentrations. Using this convention, the room temperature solubility of ZnB_3_O_4_(OH)_3_ is 0.270 wt%, comprised of 0.0267% B_2_O_3_ and 0.003% ZnO.

Polyborate species, most notably [B_5_O_6_(OH)_4_]^−^, [B_3_O_3_(OH)_4_]^−^, [B_3_O_3_(OH)_5_]^2−^, and [B_4_O_5_(OH)_4_]^2−^, exist in rapid equilibrium in relatively concentrated borate solutions with population distributions being largely a function of pH. However, at low concentrations the predominate borate species in solution are limited to monomeric B(OH)_3_ (pK_a_ = 9.2) and its conjugate base [B(OH)_4_]^−^ [11]. Since the solution resulting from zinc borate hydrolysis has a relatively low boron concentration (<0.5% boric acid eqivalent) and pH ≈ 7, B(OH)_3_ is expected to be the dominant chemical species present with only minor amounts of polyborates. Owing to the amphoteric nature of zinc, the solubilty is zinc borate is substational greatly in both acidic and basic solutions.

Concentrated aqueous slurries of zinc borate ZnB_3_O_4_(OH)_3_ are infinitely stable. A 10 wt% zinc borate suspension hydrolyzes to an extent of <4% at room temperature. Since ZnB_3_O_4_(OH)_3_ is observed to form rapidly only above about 70 °C, one might expect it to slowly convert at room temperature to another zinc borate phase, such as Zn[B_3_O_3_(OH)_5_]·H_2_O, which forms below 70 °C. However, apparently because of seeding effects, this does not occur. After continuously stirring a 10 wt% suspension of ZnB_3_O_4_(OH)_3_ at room temperature for more than one year the PXRD pattern of the suspended solids showed no indiction of the presence of other phases. This indicates that hydrolysis, at least in concentrated suspensions, is reversible. The PXRD pattern for ZnB_3_O_4_(OH)_3_ is shown in Figure 2.

When the boron and zinc concentrations of the supernatant solution in a 10 wt% aqueous suspension of zinc borate were monitored over time, it was found that approximately one month is required to reach equilibrium, as shown by the graph in Figure 3. As expected, more dilute suspensions achieve equilibrium more rapidly. A 0.05 wt% suspension of ZnB_3_O_4_(OH)_3_ completely hydrolyzes within 24 h.

ZnB_3_O_4_(OH)_3_ was found to be a stable phase when a 10 wt% aqueous suspension was maintained at the boiling point under reflux for prolonged periods of time. However, more dilute slurries were observed to convert to other phases. For example, continuous boiling under reflux of a 5 wt% aqueous suspension of ZnB_3_O_4_(OH)_3_ (2ZnO·3B_2_O_3_·3H_2_O) for one month resulted in complete conversion to the crystalline phase 6ZnO·5B_2_O_3_·3H_2_O. This is represented using oxide formulas by Equation (3) as the structural formula of 6ZnO·5B_2_O_3_·3H_2_O is not currently known. This phase was first reported by Lehmann, et al. who prepared it by heating a mixture of zinc oxide with boric acid in a 1:6–8 mole ratio with water in a sealed container for 16 h at 165 °C [12]. The formation of 6ZnO·5B_2_O_3_·3H_2_O from ZnB_3_O_4_(OH)_3_ in boiling water is preceded after one week by quantitative formation of the crystalline phase Zn_2_BO_3_(OH) (4ZnO·B_2_O_3_·H_2_O) according to Equation (4), which left in situ further converts to 6ZnO·5B_2_O_3_·3H_2_O. Methods were developed previously to manufacture Zn_2_BO_3_(OH) rapidly on large commercial scale since it is valued for its usually high dehydration onset temperature of ca. 411 °C when used as a polymer additive [13,14]. It is notable that Zn_2_BO_3_(OH) (4ZnO·B_2_O_3_·H_2_O), with B_2_O_3_/ZnO ratio 0.25, can be considered to be more hydrolyzed than 6ZnO·5B_2_O_3_·3H_2_O, with B_2_O_3_/ZnO ratio 0.83, and must reabsorb boric acid from solution to further convert to the latter.
3 2ZnO·3B_2_O_3_·3H_2_O → 6ZnO·5B_2_O_3_·3H_2_O + 8 B(OH)_3_(3)
2 ZnB_3_O_4_(OH)_3_ + 5 H_2_O → Zn_2_BO_3_(OH) + 5 B(OH)_3_(4)

Complete hydrolysis, according to Equation (2), of 100.0 g of zinc borate results in 87.2 g of boric acid and 46.7 g zinc hydroxide and requires 33.9 g of water. Thus, pure zinc borate can be considered to have a boric acid equivalence (%BAE) of 87.2% upon complete hydrolysis. This boric acid content is only released at relatively high dilution. Because of the low solubility of the Zn(OH)_2_ hydrolysis product, zinc borate does not appear to dissolve in water. This has led to occasional statements in commercial literature that zinc borate is insoluble in water. The extent of hydrolysis ZnB_3_O_4_(OH)_3_ as a function of varying suspension concentrations is illustrated in Figure 4. A 50 wt% aqueous suspension of zinc borate hydrolyzes to the extent of about 0.4%, whereas a 0.05 wt% suspension completely hydrolyzes

When the hydrolysis procedure described in Section 2.2 was carried out with a 0.10% zinc borate aqueous suspension, the solids remaining after 72 h were found to be primarily the wulfingite phase of zinc hydoxide with a minor amount of yet unhydrolyzed zinc borate, as indicated by the XRD pattern shown in Figure 5. Synthetic wulfingite, also known as ε-Zn(OH)_2_, is the most stable of the five polymorphs of Zn(OH)_2_. It has a well-defined structure composed of tetrahedral Zn^2+^ cations connected by corner sharing hydroxyl groups in the orthorhombic space group [15].

PXRD analysis of the suspended solids in a 0.05 wt% suspension of ZnB_3_O_4_(OH)_3_ after 72 h showed complete hydrolysis to Zn(OH)_2_, as shown in Figure 6. Analysis of the clear filtrate by ICP-OES indicated that it contained 77 ppm B, within error of the calculated value of 76 ppm expected for full hydrolysis of the zinc borate according to Equation (2). The zinc concentration of the filtrate, which had a pH value near 7, was found to be 10 ppm, indicating that ca. 94% of the zinc resulting from hydrolysis was in the form of insoluble Zn(OH)_2_. At this concentration and pH, it is expected that boron is primarily present as free boric acid with only a small fraction exiting in the form its conjugate base, B(OH)_4_^−^, and polyborate species.

A portion of the clear filtrate from the 0.05 wt% suspension that was filtered after 72 h was allowed to evaporate to dryness, leaving a white powder. This solid evaporite was submitted to PXRD analysis revealing the pattern of boric acid (sassolite phase) and no additional diffraction peaks, as shown in Figure 7.

## 4. Conclusions

The zinc borate, ZnB_3_O_4_(OH)_3_, known in commerce as 2ZnO·3B_2_O_3_·3.5H_2_O, is sparingly soluble in water and hydrolyzes incongruently to soluble boric acid and insoluble zinc hydroxide. The synthesis of zinc borate requires the presence of excess boric acid in the supernatant solution. When exposed to moisture in the absence of excess boric acid, hydrolysis occurs to a limited extent resulting in the release of boric acid while insoluble zinc hydroxide remains in place. This provides a mechanism for the biocidal efficacy of zinc borate, as it can act as a latent reservoir of boric acid, which becomes available as an active agent for protection only under moist conditions conducive to biodeterioration. It should be emphasized that the data presented, herein, pertain only to hydrolysis under neutral pH conditions and do not address the rate at which boric acid resulting from hydrolysis might diffuse within or deplete from manufactured articles. Building materials containing zinc borate have an excellent history of long-term durability in real-world situations [16]. Industry specific standard test methods must be used to assess potential boron depletion under relevant conditions.

## Figures and Tables

**Figure 1 molecules-27-05768-f001:**
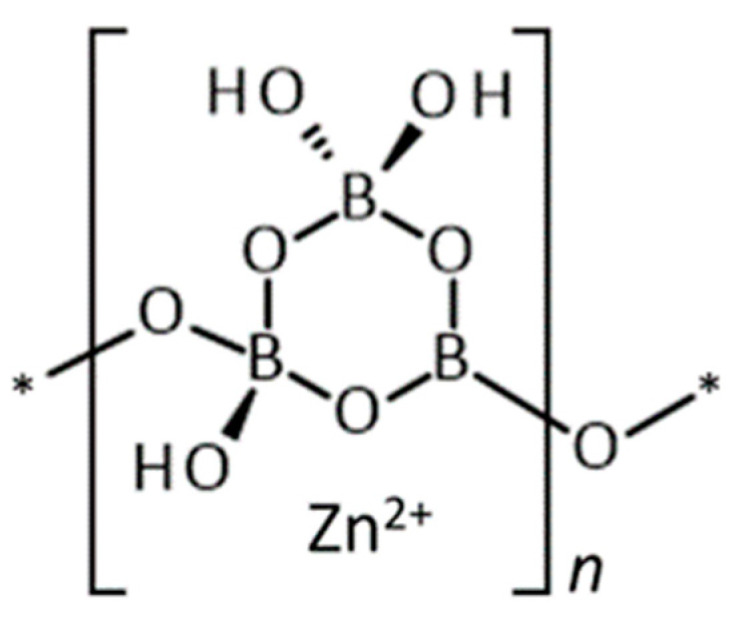
Schematic structure of ZnB_3_O_4_(OH)_3_.

**Figure 2 molecules-27-05768-f002:**
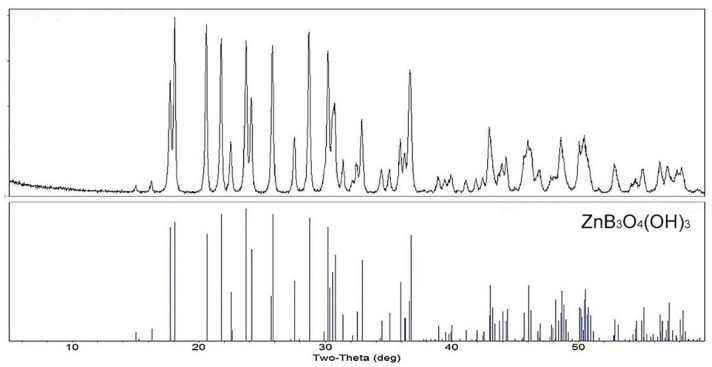
PXRD pattern of unhydrolyzed zinc borate compared to a reference file for this phase.

**Figure 3 molecules-27-05768-f003:**
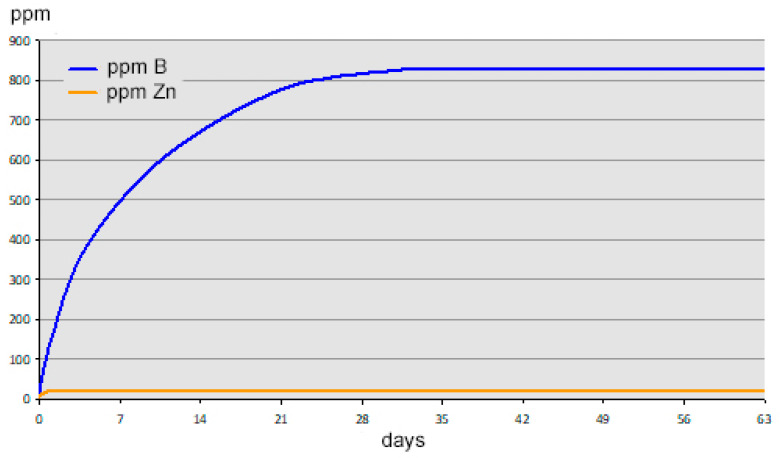
Plot of boron and zinc concentrations in solution for 10 wt% of zinc borate ZnB_3_O_4_(OH)_3_.

**Figure 4 molecules-27-05768-f004:**
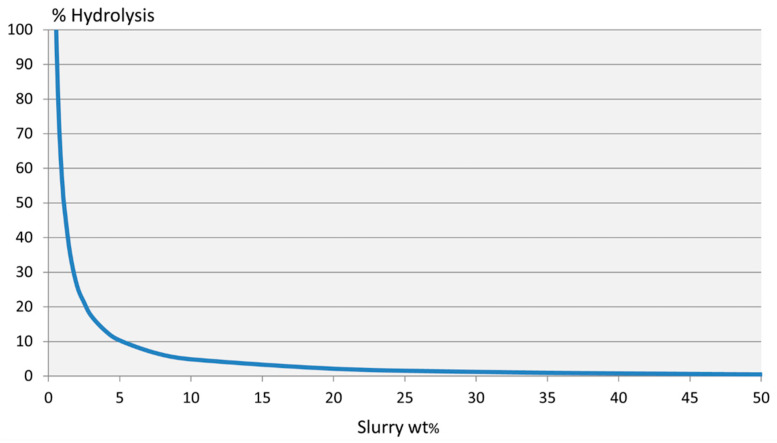
Percent hydrolysis of ZnB_3_O_4_(OH)_3_ as a function of aqueous slurry concentration.

**Figure 5 molecules-27-05768-f005:**
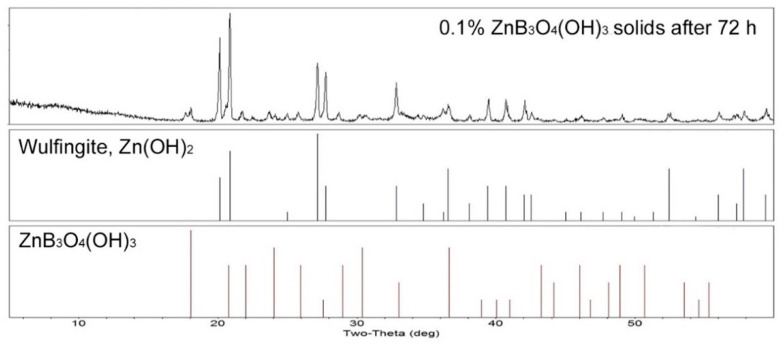
PXRD pattern of the solids resulting from hydrolysis of a 0.10% slurry of zinc borate ZnB_3_O_4_(OH)_3_ (**top**), reference peaks for wulfingite, Zn(OH)_2_ (**middle**) and zinc borate, ZnB_3_O_4_(OH)_3_ (**bottom**).

**Figure 6 molecules-27-05768-f006:**
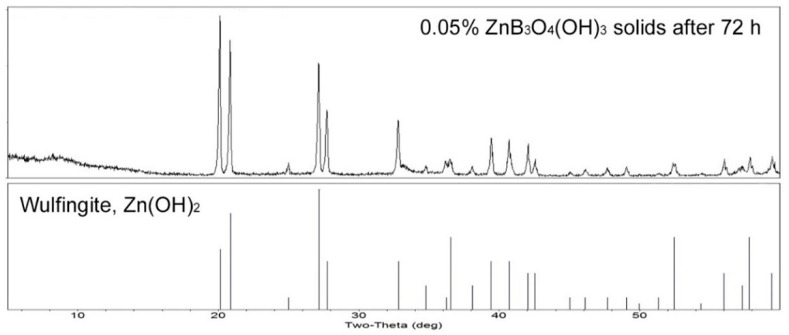
PXRD scan of the solids resulting from a 0.05% zinc borate slurry showing the wulfingite phase of Zn(OH)_2_ as the sole hydrolysis product (**top**) and reference pattern for wulfingite (**bottom**).

**Figure 7 molecules-27-05768-f007:**
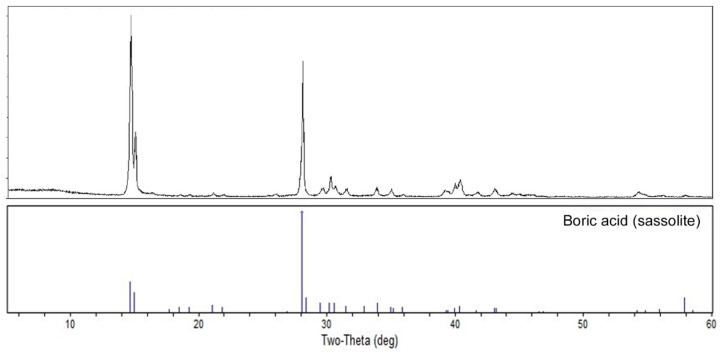
PXRD pattern of the solid evaporite obtained from the clear filtrate of a 0.05 wt% suspension of ZnB_3_O_4_(OH)_3_ after 72 h (**top**), and a reference pattern for boric acid (**bottom**).

## Data Availability

The data presented in this study are available on request from the corresponding author.

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
