# Peer review of "Zinc Borate Hydrolysis"

_molecules, 2022, doi:10.3390/molecules27185768_

Round 1

Reviewer 1 Report

In this manuscript, the hydrolysis behavior of zinc borate at neutral pH conditions was investigated. There are some problems with this manuscript.

1. The purpose of studying the hydrolysis behavior of zinc borate is unclear, and the conclusion lacks clear guiding significance.

2. The manuscript is lacking in innovation.

3. There is too little experimental data in the manuscript, and the discussion is a pale.

4. The content studied in this manuscript is of little significance for practical application. 

5. If possible, supplement the practical applications of the study. 

6. The number of references cited in this manuscript is inadequate.

Author Response

Response to Reviewer 1 Comments

Point 1: The purpose of studying the hydrolysis behavior of zinc borate is unclear, and the conclusion
lacks clear guiding significance.
Response: The purpose of studying hydrolysis is to elucidate the mechanism by which zinc
borate functions as a biocide, an applications of considerable economic importance. This is
indicated in the introduction and made particularly clear in the conclusions to the article.
Hydrolysis is also important in determining the suitability of zinc borate for use in various
applications, such as in articles that may be submerged in water for extended periods of time,
although these uses have been proposed.

Point 2: The article is lacking in innovation.
Response: Agreed. No claims are made to cutting edge science. However, this paper does
contain valuable data, interpretation, and discussion regarding an economically important boron
compound. To our knowledge, the hydrolysis behavior of zinc borate has not been previously
described in the literature.

Point 3: There is too little experimental data in the manuscript, and the discussion is a pale.
Response: This article is limited to describing the hydrolysis behavior of a single compound
which has considerable economic importance. The paper contains all necessary experimental
data to describe this behavior. Any further discussion would be superfluous. If there are other
aspects that should be addressed in the discussion to make it less pale please suggest.

Point 4. The content studied in this manuscript is of little significance for practical application.
Response: I am puzzled by this statement. As described in the introduction, this phase of zinc
borate is among the top ten boron chemicals in the world in terms industrial production an
economic importance. It is roughly number six in volume with tens of thousands of tons
produced annually. One of its main applications is as a biocide in building materials found in
many thousands of homes and commercial buildings. Its hydrolysis behavior is fundamental
to this use. Thousands of tons of zinc borate are also used as an additive in polymers and
coatings, where its hydrolysis behavior is important to informing how and where these articles
are suitable for use. It is also used as a micronutrient in agriculture, where hydrolysis is also
important. This information on relevance is included in the introduction to the paper. The
economic scale of this single phase of zinc borate is vastly greater than that of all of the
polyhedral boranes and carboranes produced in world combined. To say that this chemistry
has little significance for practical application is inaccurate.

Point 5. If possible, supplement the practical applications of the study.
Response: See response to 4.

Point 6. The number of references cited in this manuscript is inadequate.
Response: Could Reviewer 1 please suggest further relevant citations? I believe the citations
are adequate. 

Reviewer 2 Report

This is an interesting manuscript describing some unfashionable, yet important , basic science associated with an industrially significant compound.  At times it reads rather like an internal industrial report but it is well written and worthy of publication to a wider audience.  I therefore recommend publication with just a few minor changes.

I am unclear about the conclusion and the role of limited hydrolysis in the absence of excess boric acid. I am trying to work this back to the main text – I assume this is relates to line 117.  Does excess boric acid change the solubility of Zn(OH)2  e.g. line 137/138.  Perhaps this could be better explained.

Line 25. This sentence could be better worded as the citation indicates to me that [2] contains the error whereas [2] actual clears up the error.  Suggest    ….  characterization but it is now correctly formulated as the tri-hydrate [2].”

Line 71 and the example method  line  78-80 differ slightly. A version of the last sentence on line 79/80 should be added to the general method  line 71/72.

Equations and Figures should not be inserted in the middle of paragraphs but at the end of the paragraphs in which they are referred to.

Lines 136 and 228 – a citation should be added on equilibria polyborate anions systems .

Author Response

Response to Reviewer 2 Comments

Point 1: I am unclear about the conclusion and the role of limited hydrolysis in the absence of excess
boric acid. I am trying to work this back to the main text – I assume this is relates to line
117. Does excess boric acid change the solubility of Zn(OH)2 e.g. line 137/138. Perhaps this
could be better explained.

Response: I have reword this to try to make this point clearer. This phase of zinc borate can
only be synthesized in the presence of an excess of boric acid. Zinc borate hydrolyzes with
release of boric acid when exposed to water, a mechanism responsible for its efficacy as a
biocide, for example, in engineered wood building materials. The presence of boric acid does
indeed change the solubility of Zn(OH)2, which is amphoteric. In fact, any significant acid will
alter its solubility. The data reported in this paper shows a zinc concentration in solution of 25
ppm in the presence of 830 ppm B (as boric acid), which are the equilibrium concentrations
resulting from hydrolysis of zinc borate. This zinc concentration is higher than would be
obtained for Zn(OH)2 alone. In addition, the pH remains near 7 during hydrolysis, whereas it
would rise for pure Zn(OH)2. Zinc borate hydrolysis produces a buffered system, as mentioned
in the paper.

Point 2: “Line 25. This sentence could be better worded as the citation indicates to me that [2] contains
the error whereas [2] actual clears up the error. Suggest “ … characterization but it is now
correctly formulated as the tri-hydrate [2].”

Response: Thank you. I’ve made this change. My reluctance in calling it a “tri-hydrate” comes
from the fact that the compound actually contains no waters of hydration. The resolved oxide
formula factors out the three hydroxyl groups as apparent water. I’ve added some words
explaining this.

Point 3: Line 71 and the example method line 78-80 differ slightly. A version of the last sentence on
line 79/80 should be added to the general method line 71/72.
Response: Thank you. This was corrected.

Point 4: Equations and Figures should not be inserted in the middle of paragraphs but at the end of the
paragraphs in which they are referred to.
Response: Thank you. This was corrected.

Point 5. Lines 136 and 228 – a citation should be added on equilibria polyborate anions systems.
Response: A citation was added to a to polyborate anion equilibria (ref 11). Words were also
added clarifying that polyborates are not a significant aspect of this chemistry owing to a
relatively low concentrations of boron in solution (<0.5% boric acid equivalent). Only the
monomeric species B(OH)3 and [B(OH)4]
-
 are significant at the low boron concentrations
resulting from zinc borate hydrolysis. 

Reviewer 3 Report

This manuscript discusses zinc borate hydrolysis. The paper is rather short but could be interesting to chemists dealing with polyborates.

I think the manuscript could be published after adding some disussions.

1. The structure of final compound should be discussed in more detail.  I think the single-crystal X-ray diffraction data for this compound could be present in Supporting Information file and could be discussed and compared to other polyborates known in the literature. In particular, the structure of the octahedron formed in polyborates could be briefly discussed (species B3O4(OH)3, B4O5(OH)4, B5O6(OH)4, etc.). H-bonds found in the structure could be discussed. The following references could be used.

2. Please, cite refs.

https://doi.org/10.1016/j.ica.2020.119693

Please, add discussion about the structures of the compounds. 

3. As the present manuscript is dedicated to prof. Hawthorne, please, add reference to fresh boron cluster reviews

https://doi.org/10.1134/S0036023621090151 
https://doi.org/10.3390/reactions3010013

https://doi.org/10.1021/acs.jpclett.9b02290

Author Response

Response to Reviewer 3 Comments

Point 1. The structure of final compound should be discussed in more detail. I think the single-crystal Xray diffraction data for this compound could be present in Supporting Information file and could
be discussed and compared to other polyborates known in the literature. In particular, the
structure of the octahedron formed in polyborates could be briefly discussed (species
B3O4(OH)3, B4O5(OH)4, B5O6(OH)4, etc.). H-bonds found in the structure could be
discussed. The following references could be used.

 Point 2. Please, cite refs.
 https://doi.org/10.1016/j.ica.2020.119693
Please, add discussion about the structures of the compounds.

Response: I’ve added details and a reference on polyborate anions as well as on the structure
of -Zn(OH)2, the hydrolysis product formed along with boric acid. I’ve also added more details
on the structure of the phase of zinc borate. Full details can be found in reference 2.
This study involved only powder X-ray diffraction to identify phases. No single-crystal or
Rietveld structural characterization was done, nor would it be warranted since the structure of
the subject zinc borate has already been reported (by us, ref 2) and the hydrolysis products are
simply boric acid and -Zn(OH)2, both of which have well-known structures described in the
literature. Nevertheless, I’ve added some discussion about these structures.
Polyborate anions only form in relatively concentrated solutions of boric acid (>0.5%) in the
intermediate pH regime. Polyborates form a huge variety of metal complexes for which I’ve
provided an excellent reference (ref 10, Beckett). I have published many papers on the
synthesis of metal and non-meal polyborate compounds myself which I did not cite in this paper
because polyborates are not relevant as hydrolysis products. Hydrolysis of sparingly soluble
zinc borate leads to a dilute boron-containing solution in which polyborate formation is not
significant. In dilute solution, monomeric B(OH)3 and its conjugate base, [B(OH)4]-, are the only
significant species present. I’ve added some language to the paper making this point. The
suggested reference describes the decomposition of a polyhedral borane anion and
concomitant formation of metal complexes, which is not relevant to the hydrolysis of zinc borate
which yields neither polyborates nor their metal complexes.

Point 3: As the present manuscript is dedicated to prof. Hawthorne, please, add reference to fresh
 boron cluster reviews”
https://doi.org/10.1134/S0036023621090151
https://doi.org/10.3390/reactions3010013

Response: When I was invited to submit an article to this special issues I believed the topic
was boron chemistry without restriction to polyhedral boranes. I was motivated to submit to this
issue because of my long association with Prof. Hawthorne. I served as his head postdoc,
which he called his “Group Lieutenant”, for nearly six years. In this role I had the privilege to
discuss chemistry with him on a daily basis, help manage his world-class UCLA research
group, and conduct extensive hands-on laboratory work on metallacarboranes. I co-authored
numerous papers with Hawthorne and, after leaving his group, maintained a lifelong
association with him. Thus, I likely have a stronger affiliation with the honoree of this issue than
any other contributor. While my field of boron research has shifted to the more industrially
relevant boron oxides, I know that Fred Hawthorne loved all aspects of boron chemistry, and I
believe he would approve. I note that several papers already published in this special issue do
not involve boron hydrides clusters. Nevertheless, if the editors feel this article does not fit with
this special issue I will submit it elsewhere.
Since my article deals entirely with boron oxide compounds and not at all with boron hydride
clusters, I do not see how the suggested citations are relevant or appropriate to include in the
paper. 

Round 2

Reviewer 1 Report

The author has made the necessary modifications according to the comments of the reviewers, and it is ready for publication.

Reviewer 3 Report

The paper can be published in the present form.